# BRAF^V600E^ Induction in Thyrocytes Triggers Important Changes in the miRNAs Content and the Populations of Extracellular Vesicles Released in Thyroid Tumor Microenvironment

**DOI:** 10.3390/biomedicines10040755

**Published:** 2022-03-23

**Authors:** Ophélie Delcorte, Catherine Spourquet, Pascale Lemoine, Jonathan Degosserie, Patrick Van Der Smissen, Nicolas Dauguet, Axelle Loriot, Jeffrey A. Knauf, Laurent Gatto, Etienne Marbaix, James A. Fagin, Christophe E. Pierreux

**Affiliations:** 1CELL Unit, de Duve Institute, Université Catholique de Louvain, 1200 Brussels, Belgium; catherine.spourquet@uclouvain.be (C.S.); pascale.lemoine@uclouvain.be (P.L.); jonathan.degosserie@uclouvain.be (J.D.); patrick.vandersmissen@uclouvain.be (P.V.D.S.); etienne.marbaix@uclouvain.be (E.M.); 2PICT Platform, de Duve Institute, Université Catholique de Louvain, 1200 Brussels, Belgium; 3CYTF Platform, de Duve Institute, Université Catholique de Louvain, 1200 Brussels, Belgium; nicolas.dauguet@uclouvain.be; 4Computational Biology and Bioinformatics, de Duve Institute, Université Catholique de Louvain, 1200 Brussels, Belgium; axelle.loriot@uclouvain.be (A.L.); laurent.gatto@uclouvain.be (L.G.); 5Department of Medicine and Human Oncology & Pathogenesis Program, Memorial Sloan Kettering Cancer Center, New York, NY 10065, USA; knaufj@ccf.org (J.A.K.); faginj@mskcc.org (J.A.F.)

**Keywords:** thyroid cancer, PTC, BRAF^V600E^, mouse model, miRNA, extracellular vesicles, heterogeneity, sequencing

## Abstract

Papillary thyroid cancer (PTC) is the most common endocrine malignancy for which diagnosis and recurrences still challenge clinicians. New perspectives to overcome these issues could come from the study of extracellular vesicle (EV) populations and content. Here, we aimed to elucidate the heterogeneity of EVs circulating in the tumor and the changes in their microRNA content during cancer progression. Using a mouse model expressing BRAF^V600E^, we isolated and characterized EVs from thyroid tissue by ultracentrifugations and elucidated their microRNA content by small RNA sequencing. The cellular origin of EVs was investigated by ExoView and that of deregulated EV-microRNA by qPCR on FACS-sorted cell populations. We found that PTC released more EVs bearing epithelial and immune markers, as compared to the healthy thyroid, so that changes in EV-microRNAs abundance were mainly due to their deregulated expression in thyrocytes. Altogether, our work provides a full description of in vivo-derived EVs produced by, and within, normal and cancerous thyroid. We elucidated the global EV-microRNAs signature, the dynamic loading of microRNAs in EVs upon BRAF^V600E^ induction, and their cellular origin. Finally, we propose that thyroid tumor-derived EV-microRNAs could support the establishment of a permissive immune microenvironment.

## 1. Introduction

Thyroid cancer, with papillary thyroid carcinoma (PTC) accounting for 85% of cases, is the most common endocrine malignancy [1] and the most rapidly increasing cancer in the US [2]. Despite the high survival rate of patients [3], clinicians are facing two challenges. First, the overdiagnosis of nodules or indolent microcarcinomas leads to overtreatment and unnecessary surgeries [4,5]. Second, 10–20% of patients present aggressive, poorly differentiated, metastatic thyroid tumors often associated with BRAF^V600E^ mutation and poor prognosis [6,7,8]. Continued fundamental research to improve differential diagnosis between thyroid cancer stages and types and to better understand the biology of PTC is thus required. For those reasons, attention have recently been drawn on extracellular vesicles (EVs) [9,10,11].

EVs are small lipid bilayer-enclosed vesicles released by virtually all cell types. EVs transport a specific cargo including proteins and nucleic acids that constitutes a complex signal. This signal can be transferred to a receiver cell supporting local and/or distant communication in several physiological and pathological processes, most noticeably cancer [12,13,14]. Among the different component of EV cargo, microRNAs (miRNAs) attracted a lot of interest for their functional versatility. MiRNAs are endogenous single-stranded non-coding RNA molecules ranging from 18 to 22 nucleotides in length, which inhibit the translation of target mRNAs. Deregulation of miRNAs expression has been associated with the onset, progression and metastasis of various cancers [15,16]. EV-miRNAs secreted by tumor cells are now considered important actors in cancer progression [17], and as promising non-invasive markers for disease monitoring [11,18].

Although many studies have demonstrated the EV-mediated transfer of biological molecules, including miRNAs, to recipient cells, the origin of these EVs has been mostly derived from homogeneous cultured cell lines [14,19,20,21]. However, the histological complexity of tumors is such that multiple cell types produce EVs whose origins, properties and effects on the tumor microenvironment (TME) and progression are still largely unexplored. Only a few studies have recently examined the properties of tissue-derived EVs, including EVs directly isolated from tumors [22,23,24,25]. For example, Cianciaruso et al. demonstrated that macrophages-derived EVs represented approximately 60% of EVs isolated from MC38 tumors, thereby stressing the impact of cellular populations present in the microenvironment, and their EV production potential in the EV pool [23].

In this study, we aimed to elucidate the heterogeneity of EV populations within the thyroid tumor, a complex tissue, as well as the dynamic of miRNAs expression and loading in extracellular vesicles during cancer progression. We used a genetically engineered mouse model of PTC in which BRAF^V600E^ expression can be selectively induced in thyrocytes [26]. This model allowed to harvest the EVs released by thyroid tissue, to investigate their cellular origin, to profile the evolution of miRNA expression in tissue and their loading in EVs during BRAF^V600E^-dependent PTC development. We integrated miRNA deregulation in the different cell types of the TME with the changes in EV populations produced within the tissue. In silico analyses revealed that the identified EV-miRNAs could be involved in immune cell modulation within the tissue. Altogether, our results provide insights into the dynamic of miRNA deregulation in PTC upon BRAF^V600E^ induction and identify EV-miRNAs that could play a role as actors or markers of PTC development. Our results shed light on intercellular communications by EVs occurring in a complex developing thyroid tumor.

## 2. Materials and Methods

We submitted all the relevant data from our experiments to the EV-TRACK knowledgebase (EV-TRACK ID: EV210002) [27].

### 2.1. Mice

Tg-rtTA and tetO-BRAF^V600E^ mice were obtained from J. A. Fagin [26]. Single homozygous mice in the FVB background were crossed to generate double heterozygous Tg-rtTA/tetO-BRAF^V600E^ mice in which the oncogene BRAF^V600E^ can be selectively expressed in thyrocytes upon doxycycline administration. Doxycycline was delivered to mice at 10–14 weeks of age via intraperitoneal injection (1 µg/g body weight) every 24 h for 24 (day 1), 48 (day 2), 72 (day 3) or 96 h (day 4). Control mice were injected with saline sodium chloride solution.

### 2.2. Tissue Collection and Histology

Mice were sacrificed by cervical dislocation under anesthesia by 150 µL of a xylazine (20 mg/kg)/ketamine (200 mg/kg) solution and after cardiac puncture. Thyroid lobes were excised and either snap-frozen for RNA analysis or cut in 1 mm^3^ pieces for tissue dissociation and EV isolation. For histological and immunohistochemical analyses, thyroid lobes were extracted together with trachea, fixed in 4% paraformaldehyde and embedded in paraffin using a Tissue-Tek VIP-6 (Sakura, Torrance, CA, USA). Sections of 6 µm were obtained with the microtome Micron HM355S (ThermoScientific, Waltham, MA, USA). After paraffin removal and rehydration, histological sections were stained with hematoxylin and eosin, slides were mounted in Dako aqueous Medium (Agilent Technologies, Santa Clara, CA, USA) and scanned with the panoramic P250 digital slide scanner (3DHistech, Budapest, Hungary), as described in [28].

### 2.3. Immunohistochemistry and Immunohistofluorescence

After paraffin removal and endogenous peroxidases inhibition with 0.3% H_2_O_2_ for immunohistochemistry, the slides were treated as described in [29] for antigen retrieval, permeabilization and the blocking of non-specific sites. Anti-phospho-ERK primary antibody (Cell Signaling Technology, Leiden, The Netherlands, #4376; 1/200) was incubated overnight at 4 °C. HRP-conjugated secondary antibody was incubated in PBS/10% BSA/0.3% Triton X-100 for 1 h at room temperature (RT). Diaminobenzidine was used for the revelation of phospho-ERK immunostained cells while hematoxylin staining allowed the visualization of tissue structure. Finally, slides were mounted in Dako aqueous medium (Agilent Technologies, Santa Clara, CA, USA) and scanned with the panoramic P250 digital slide scanner (3DHistech, Budapest, Hungary). For immunohistofluorescence, thyroid lobes were embedded in gelatin and frozen sections of 6 µm were obtained with the cryostat (ThermoScientific, Waltham, MA, USA), as described in [30]. Anti-CD206 primary antibody was incubated overnight at 4 °C. Secondary antibody coupled to Alexa-488 (Invitrogen, Waltham, MA, USA) and fluorescent nuclear dye (Hoechst 33258; Merck, Kenilworth, NY, USA) were used. Slides were observed with the Zeiss Cell Observer Spinning Disk confocal microscope.

### 2.4. mRNA Quantification

Total RNA was extracted from thyroid lobes or bone marrow derived macrophages (BMDM) using TRIzol Reagent (ThermoScientific, Waltham, MA, USA), as described in [28], and followed by a second phenol/chloroform purification step. Reverse transcription was performed on 500 ng of total RNA using M-MLV reverse transcriptase (Invitrogen, Waltham, MA, USA) and random hexamers. Real-time quantitative PCR was performed as described in [31]. Primers sequences are listed in Appendix A. Data were analyzed using the ΔΔCT method, with the geometric mean of *Gapdh* and *Rpl27* expression as reference.

### 2.5. Flow Cytometry and FACS

Thyroid lobes were cut in 1 mm^3^ pieces and incubated in dissociation solution (DMEM Ca^2+^ free, Collagenase P 1 mg/mL, Dispase II 2.5 mg/mL, DNase 5 U/mL) under stirring for 40 min at 37 °C with homogenization using a 1 mL micropipette every 8 min. Then, cell suspension was passed through a 40 µm-filter and diluted in DMEM Ca^2+^ free, FBS 20%, 1 mM EDTA pH8 to stop dissociation. Cells were centrifuged and resuspended in blocking solution (PBS, 1 mM EDTA, FBS 1%, 1 µg TrueStain FcXTM, #101320, Biolegend, San Diego, CA, USA) for 10 min. The identification of immune populations by flow cytometry was performed using anti-CD45, anti-CD11b and anti-CD3 fluorochrome-conjugated antibodies. Dead cells were visualized by DAPI staining. Data were acquired on the BD LSRFortessa™ Cell Analyzer and analyzed with FlowJoTM software (BD Biosciences, Franklin Lakes, NJ, USA). FACS experiments were performed on the BD FACSAria™ III Cell Sorter. Cells were sorted using anti-E-cadherin (Ecad), anti-CD31 and anti-CD45 fluorochrome-conjugated antibodies, and recovered in TRIzol reagent. The triple negative population was also collected. All antibodies and working conditions are recapitulated in Appendix A.

### 2.6. Tissue-EVs Isolation

For tissue dissociation and EV isolation, all solutions were filtrated on a 0.2 µm-filter and each step was performed at 4 °C. Thyroid lobes were microdissected, cut in 1 mm^3^ pieces and incubated overnight under stirring in Trypsin-EDTA 0.05% (Life Technologies, Carlsbad, CA, USA). After homogenization with a 1 mL micropipette, trypsin-EDTA supernatant was diluted with BSA 5% and conserved at 4 °C. Non-dissociated tissue pieces were incubated in a collagenase type II solution (400 U/mL, LS004176, Worthington, Columbus, OH, USA) for 60 min. Enzymatic dissociation was complemented with mechanical dissociation by passing the solution through syringes of decreasing diameter (0.8 µm; 0.6 µm; 0.5 µm). The resulting cell suspension was finally recovered with the first trypsin-EDTA supernatant in BSA 5% and submitted to a 2000× *g* centrifugation for 15 min (GR 4.11, Jouan) to pellet dissociated cells. For the analysis of EV-miRNAs, the supernatant was treated with RNase A (ThermoScientific, Waltham, MA, USA) at 10 µg/mL for 45 min at 37 °C prior to differential ultracentrifugation as described in [32]: 20,000× g for 15 min in a Ti50 rotor (Beckman, Brea, USA), and 150,000× *g* for 90 min in a Ti80 rotor (Beckman, Brea, CA, USA). This last 150k pellet was washed once by suspension in PBS and resedimentation at 150,000× *g* for 90 min. This material is hereafter referred to as “EV-pellet”. All the procedure is schematized in Appendix A.

### 2.7. Tissue-EVs Characterization

Particles in EV-pellets were counted with the Zetaview (Particle Metrix GmbH, Inning am Ammersee, Germany), as described in [33]. Briefly, the diluted samples in pre-filtered PBS (0.2 µm filter) were analyzed with a sensitivity of 85 and a camera shutter of 100. Measurements were performed with a medium resolution of two cycles at 11 different positions (or frames) of the cell chamber containing the sample.

EV-pellets were observed by scanning electron microscopy (SEM), as described in [33]. For transmission electron microscopy (TEM) after negative staining, all samples were appropriately diluted in water with a final concentration of 1% uranyl acetate stain. A 50 µL drop of diluted sample and stain was put on formvar and carbon-coated grid for 5 min; then, the excess of liquid was removed by filter paper, and the specimen was dried overnight in a closed box containing dried silica gel. TEM and SEM samples were observed in a transmission electron microscope (CM12, Philips, Amsterdam, The Netherlands) at 80 kV.

Western blotting was performed as described in [33]. EV pellets were lysed in RIPA buffer, and protein concentration was measured using a bicinchoninic acid assay. EV-pellet lysates, normalized to protein content, were resolved on 8 or 10% homemade polyacrylamide gels, and transferred onto the PVDF membrane. These were blocked in TBS-Tween20 0.05%-milk 5% and incubated overnight with primary antibodies listed in Appendix A. HRP-conjugated secondary antibodies were incubated in TBST-milk 0.5% for 1 h. Immuno-reactive bands were detected using Super Signal Chemiluminescent Substrate (ThermoScientific, Waltham, MA, USA) and images were acquired using Fusion Solo S (Vilber Lourmat, Collégien, France).

EV-pellets were analyzed using the ExoView Mouse Tetraspanin Kit (NanoView Biosciences, Boston, MA, USA). Samples were diluted in manufacturer-supplied incubation solution and incubated overnight at RT on ExoView Mouse Tetraspanin Chips. After washing, chips were incubated with anti-CD9, anti-CD81, anti-CD11b, anti-CD31, anti-CD45 and anti-Ecad fluorochrome-conjugated antibodies (listed in Appendix A) for 1 h at RT with shaking. Chips were then washed, dried and imaged by the ExoView R100 using ExoView Scanner v3.1. Data were analyzed using ExoView Analyzer v3.1. Fluorescent cut-offs were set relative to the Rat IgG and Hamster IgG controls.

### 2.8. MiRNA Quantification

Total RNA was extracted from RNase-treated EV-pellets and from dissociated tissues using miRNeasy Mini kit (QIAGEN, Hilden, Germany) according to the manufacturer’s instructions. RNA was resuspended in 30 µL of RNase-free water and stored at −80 °C. RNA was quantified using Quant-iT™ RiboGreen™ RNA Assay Kit (ThermoScientific, Waltham, MA, USA). Pulsed reverse transcription was performed on 10 ng of total RNA using M-MLV Reverse Transcriptase (Invitrogen, Waltham, MA, USA) and a mix of 5 nM specific stem-loop primers, as described in [32]. Real-time quantitative PCR was performed as described in [34], in the presence of 300 nM of universal reverse primer, 300 nM of the specific forward primer, 250 nM of probe with Takyon Probe Master Mix (Eurogentec, Liège, Belgium) on a CFX96 touch real-time PCR Detection System (Bio-Rad, Hercules, CA, USA). Primers and probes sequences are listed in Appendix A. Data were analyzed using the ΔΔCT method, with let-7c-5p or the geometric mean of miR-126a-3p and let-7b-5p as reference miRNAs for cellular samples or EV samples, respectively.

### 2.9. MiRNA Sequencing

Total RNA from EV-pellets and dissociated tissues were sent to the QIAGEN RNA sequencing platform (Hilden, Germany) for miRNA next-generation sequencing. The library was prepared using Qiaseq miRNA NGS kit and the Illumina platform was used for sequencing. Filtered reads were annotated using miRbase v22, with reference genome GRCm38. All analyses were carried out using CLC Genomics Workbench (version 20.0.2) and CLC Genomics Server (version 20.0.2) by QIAGEN. KEGG pathway analysis was performed with the DIANA mirPath v.3 web server using microT-CDS v.5 as prediction tool [35].

### 2.10. Statistical Analysis

All graphs and statistical analyses were performed with the Prism software (GraphPad Software). Real-time qPCR values were obtained by the ΔΔCT method and are expressed as mean  ±  standard deviation (SD) [36]. Each graph represents the results from a minimum of four independent experiments. Nonparametric statistical tests were used: Kruskal–Wallis followed by Dunn’s post-test for multiple comparisons. Differences were considered statistically significant when *p* < 0.05 (*); # stands for *p* < 0.01; $ for *p* < 0.001. Supplementary methods can be found in Appendix B.

## 3. Results

### 3.1. BRAF^V600E^ Expression in Mouse Thyrocytes Triggers Progressive Thyroid Tissue Transformation Mimicking Features of Human PTC

To trigger BRAF^V600E^ expression in thyrocytes, we treated Tg-rtTA/tetO-BRAF^V600E^ mice by the intraperitoneal injections of doxycycline every 24 h for 2 or 4 days. Macroscopically, the thyroid size was normal after 2 days (i.e., after injections at 0 h and 24 h and analysis at 48 h), but clearly enlarged at 4 days (Appendix A). Histologically, the loss of follicular structure, colloid consumption and papillae formation were initiated at 2 days (Figure 1A). In addition, a strong stromal and inflammatory reaction was observed at 4 days (Figure 1A). At the cellular level, transformed thyrocytes exhibited an irregular shape, multilayering and increased ERK phosphorylation (Figure 1B). The expression of MAPK pathway target genes, *Fosl1* and *Dusp5*, was dramatically increased (Figure 1C). Conversely, the expression of thyroid-specific genes (*Nis*, *Tpo*, *Tg* and *Tshr*) and of thyroid transcription factors (*Pax8* and *Nkx2.1*) were decreased after 2 days and 4 days, indicating progressive thyroid dedifferentiation (Figure 1D). MiRNA deregulation was assessed by measuring the expression of some well-known miRNA candidates identified in human PTC. We found that the expression of miR-31-5p, miR-222-3p, miR-146b-5p and miR-21a-5p was significantly increased as early as 2 days of injections (Figure 1E). On the contrary, the expression of miR-26a-5p, miR-148a-5p, miR-30d-5p, miR-99a-5p and miR-10b-5p was only significantly decreased after 4 days (Figure 1E).

Altogether, histological and molecular analyses demonstrated that the induction of BRAF^V600E^ expression in thyrocytes triggered rapid and evident morphological changes in the entire thyroid, over-activation of MAPK pathway, thyrocytes dedifferentiation and then deregulation of miRNA expression. Those features recapitulated events that occur in human PTC, proving the relevance of this mouse model to study the early events triggered by BRAF^V600E^ expression.

### 3.2. Control and BRAF^V600E^ Thyroids Release EVs with Exosomal Characteristics

To test whether thyroid tissue directly releases extracellular vesicles, we optimized a gentle dissociation protocol of microdissected control and BRAF^V600E^-transformed thyroid lobes combined with differential ultracentrifugations (Appendix A). Particles in the high-speed pellet, supposedly enriched in EVs, were measured by nanoparticle tracking analysis (NTA). Analysis revealed a similar size distribution of particles in the EV-pellets from the control, day 2 and day 4 samples, with a median diameter between 130 and 160 nm and a mean diameter at approximately 200 nm (Figure 2A). The particle size and shape were also determined by scanning and transmission electron microscopy. Electron microscopy images showed isolated double-membrane vesicles of approximately 50–150 nm in diameter (Figure 2B). NTA further revealed increased particle counts upon BRAF^V600E^ expression (Figure 2C). Finally, western blotting demonstrated the presence of several EV-specific markers in the high-speed pellet with an enrichment of CD63, CD9, CD81, Alix and Flotilin-1 as compared to total cellular lysates (Figure 2D). Conversely, the absence of two reticulum endoplasmic markers (Calnexin and Protein disulfide-isomerase) in the EV-pellets argued against contamination by intracellular membrane compartments (Figure 2D). We concluded that control- and BRAF^V600E^-thyroid release extracellular vesicles with exosomal characteristics.

### 3.3. BRAF^V600E^ Induction Affects the EV Populations Found within the Thyroid

To evaluate the heterogeneity of EV populations within thyroid tissue and unravel the intercellular crosstalk that could be triggered within the tumor microenvironment, we aimed to identify the cellular origin of thyroid- and tumor-derived EVs. We analyzed the presence of cell population markers on the EV surface with the ExoView technique. Capture antibodies were directed against CD9 and CD81, two tetraspanins previously detected by western blot (Figure 2D). Fluorochrome-conjugated antibodies were directed against specific cell population markers, Ecad, CD31, CD45 and CD11b to identify the EVs produced by epithelial cells, endothelial cells and immune cells. E-cadherin+ EVs were CD9+ and/or CD81+ (Figure 3A). In contrast, CD31+ and CD11b+ EVs were mostly CD9+ (Figure 3A). There was negligible (<1.4%) Ecad/CD31 colocalization validating the specificity of the technique (Appendix A). Overall, endothelial CD31+ and immune CD11b+ EVs were the most abundant populations, respectively representing 31 and 56% of total CTL-EVs, and 23 and 49% of PTC-EVs pool (Figure 3B). In PTC tissue compared to normal tissue, all EV populations were increased (Figure 3A, red bars), thereby reflecting the overall increase in EV count (Figure 2B). More importantly, Ecad+ EVs and CD45+ EVs were drastically and proportionally more abundant in PTC tissue as compared to normal tissue (Figure 3A,B). Finally, Ecad fluorescence intensity was increased on CD81+ PTC-EVs, indicating an increase in Ecad molecules per EV in PTC tissues (Appendix A). Similarly, CD45 fluorescence intensity was increased on CD9+ and CD81+ PTC-EVs compared to CTL-EVs (Appendix A). The size measurement of the CD81+ population showed a shift of the mean diameter between CTL-EVs and PTC-EVs from 60.3 nm to 71.7 nm while the mean diameter of the CD9+ population was stable (Appendix A). This confirmed changes in the CD81+ EV population. In conclusion, Ecad+ EVs and CD45+ EVs, respectively, derived from epithelial cells and immune cells, constituted a minor fraction (11%, Figure 3B) of thyroid tissue-derived CD9+/CD81+ EV pool, but showed a doubling in their proportion within the total EV pool of tumor tissue (23%, Figure 3B).

Cell sorting based on Ecad, CD31 and CD45 markers revealed that these three populations were increased in PTC-tissues, supporting tissue growth (Figure 3C and Appendix A). Immune CD45+ cells displayed the most drastic, 40-fold, increase in cancerous tissue, in which they accounted for 50% of non-epithelial cells (Figure 3C,D). Endothelial cells represented the most abundant non-epithelial cell type in normal tissue and the less abundant one in cancerous tissue (Figure 3D), despite a two-fold increase (Figure 3C). It is important to mention that a large fraction of epithelial cells remained on the filter due to incomplete epithelial cell dissociation. Nevertheless, the doubling observed by FACS (Figure 3C) illustrated the increased number of tumor cells observed on histological sections (Figure 1A). In conclusion, we identified two deregulated EV populations in the thyroid tumor, CD45+ EVs and Ecad+ EVs. While the increased abundance of CD45+ EVs probably represented the increase in CD45+ cells (Figure 3C,D), the increased abundance of Ecad+ EVs (approximately 10-fold) was more important than the expansion of the epithelial tissue (Figure 1A and Figure 3C). This suggested that BRAF^V600E^-transformed cells would produce more EVs than normal thyroid cells. 

### 3.4. BRAF^V600E^ Induction Triggers Differential and Progressive miRNA Loading in Thyroid Tissue EVs

We then focused on the RNA content of EVs isolated from thyroid tissue. The electrophoretic profiles of total RNA from CTL- and PTC-EVs (day 2 and day 4) were similar, lacked a tRNA peak, arguing against artefactual intracellular contamination (Appendix A), and showed a predominance of small RNA species with a peak at approximately 20 nucleotides, most probably corresponding to miRNAs (Appendix A). In addition, RNA quantification revealed that the total RNA content of particles did not significantly change upon the induction of BRAF^V600E^ expression (Appendix A). After floatation in a density gradient (Appendix A), miRNAs were shown to be associated with EV-fractions (Appendix A: high number of particles, CD9 signal, density of 1.08–1.15).

To elucidate the miRNA content of EVs released by thyroid tissue, we sequenced the miRNAs of EVs isolated from control- and BRAF^V600E^-thyroids (day 2 and day 4). The hierarchical clustering of the samples using miRNA expression patterns allowed the separation of EV samples according to the treatment, revealing dynamic and progressive changes in miRNA abundance in EVs upon BRAF^V600E^ induction (Figure 4A), as observed in tissues (Figure 1E). Differential expression analysis identified 26 and 97 miRNAs with differential abundance (DA) between the control- and PTC-EVs at 2 and 4 days, respectively (FDR *p*-value < 0.05). The volcano-plot showed that DA-miRNAs in day 2 samples were predominantly more abundant in EVs upon BRAF^V600E^ expression (Figure 4B,D), supporting the rapid upregulation of miRNAs observed in tissues (Figure 1E). After 4 days of BRAF^V600E^ induction, the 97 DA-miRNAs were equally distributed in the plot, indicating a similar number of more abundant and less abundant miRNAs in EVs (Figure 4C,D).

To confirm sequencing data and establish a time-course of miRNA deregulation in tissue and EVs at early stages of PTC development, miRNAs levels were quantified by RT-qPCR in dissociated tissues (Appendix A) and in EVs (Appendix A) after 1, 2, 3 and 4 days of doxycycline injections. The results revealed the upregulation of miR-31-5p, miR-146b-5p, miR-21a-5p, miR-221-3p and miR-222-3p in tissue and a higher abundance in EVs as early as day 1 of doxycycline treatment (Figure 4E), when tissue structure and differentiation started to be impacted (Figure 1). EV-miRNAs presenting a reduced abundance at sequencing after BRAF^V600E^ induction showed a trend towards reduced abundance at day 3 and day 4 in tissue as in EVs (Appendix A). Altogether, these results pinpoint a progressive and differential deregulation of EV-loaded miRNAs that mirror the changes occurring in BRAF^V600E^ thyroid tissues: a rapid and prolonged upregulation of miRNAs with a delayed downregulation of other miRNAs. 

### 3.5. MiRNAs More Abundant in Tumor-Derived EVs Are Mainly Produced by Epithelial Cells

We then investigated which cell populations were responsible for the increased miRNA abundance in EVs. For this, we assumed that the population with the highest expression of a miRNA of interest would be the main source of EVs containing this miRNA. We selected four miRNAs, miR-31-5p, miR-21-5p, miR-146b-5p and miR-221-3p, as the most abundant and rapidly increasing EV-miRNAs in BRAF^V600E^-expressing thyroid tissues (Figure 5A,B). We measured their expression in FACS-sorted epithelial (Ecad+), endothelial (CD31+), immune (CD45+) and mesenchymal (Ecad─CD31─CD45─) cells from CTL- and PTC-tissues (Appendix A). The purity of sorting was assessed by measuring the enrichment of some canonical epithelial (Appendix A), endothelial (Appendix A), immune (Appendix A) and mesenchymal (Appendix A) genes in each population. The expression of the four miRNA candidates were only or mainly increased in the epithelial population (Figure 5C, violin plots). Moreover, in PTC tissues, they were mostly expressed by this population (Figure 5C, stacked bars at right). Importantly, their expression in the immune population was constant or decreased (Figure 5C, violin plots). To consider the changes in proportion of each cell population between the control and BRAF^V600E^-thyroids (Figure 3C), we adjusted the miRNA relative expression to the amount of RNA extracted by non-epithelial cells. Despite an important increase in immune cells’ number, the expression by the immune compartment was still representing only a negligible part of the total expression by non-epithelial cells (Appendix A).

In summary, the number of epithelial cells and of epithelial EVs was increased in PTC tissue compared to the control tissue (Appendix A and Figure 3A), and the more abundant EV-miRNAs were mainly expressed and deregulated in the epithelial cells of PTC tissue (Figure 5C). From those data, we concluded that the deregulation in miRNA cargo that we measured in the total PTC-derived EV pool was mainly due to epithelial/thyrocyte-EVs.

### 3.6. MiRNAs More Abundant in Tumor-Derived EVs Could Be Involved in Immune Cells Regulation

To propose a role for miRNAs loaded in EVs, we focused on miRNAs specifically enriched in PTC-derived EVs. Indeed, these were the first deregulated upon BRAF^V600E^ expression (Figure 4B,D) and the most susceptible to be implicated in intercellular communication via EVs within the tumor microenvironment. The KEGG pathway analyses of miRNAs showing an increased abundance in EVs isolated from BRAF^V600E^ thyroid tissues revealed an association with several pathways related to cancer and cytokine-cytokine receptor interaction (Figure 6A,B for day 2 and day 4). Since upregulation in EV-miRNAs species were mainly caused by epithelial cells, these in silico analyses led us to consider a potential crosstalk between the tumor and immune cells and to characterize the changes occurring in the immune compartment after BRAF^V600E^ induction in thyrocytes.

Flow cytometry experiments were performed to identify the immune populations among dissociated control- and BRAF^V600E^-thyroids. CD45-positive cells were dramatically increased in BRAF^V600E^-thyroids (Figure 3C), with CD11b-positive myeloid cells accounting for 20–30% of CD45+ cells (Figure 6C,D). Alternatively-activated macrophages contributed to the increase in the number of myeloid cells (Figure 6E), as shown by the immunohistofluorescence for CD206 (Figure 6F). Altogether, those results demonstrated important changes in the immune compartment after BRAF^V600E^ induction in thyrocytes, with the recruitment of anti-inflammatory pro-tumoral macrophages, most likely creating a permissive immune microenvironment for cancer progression.

Despite the low abundance of Ecad+ EVs in the total pool of EVs (Figure 3B), we tested whether EV-miRNAs released by BRAF^V600E^ thyroid tumors could affect or polarize the primary culture of bone marrow-derived macrophages (BMDMs). PKH67-labeled PTC-EVs were readily internalized by macrophages (Appendix A). However, we could not demonstrate reproducible effects of PTC-EVs by transcriptomic analysis. In two out of the three independent experiments, we showed that CTL-EVs induced genes involved in immune and inflammatory-signaling pathways (Appendix A). BMDMs treated with CTL-EVs shared a pro-inflammatory gene signature with in vitro (LPS + IFNγ; Appendix A) or in vivo (LPS+; Appendix A) classically activated macrophages. On the contrary, PTC-EVs induced genes were implicated in cell cycle and proliferation (Appendix A). In addition, BMDMs treated with PTC-EVs shared an anti-inflammatory gene signature with in vitro (IL-4; Appendix A) or in vivo (LPS-; Appendix A) alternatively activated macrophages. However, the expression of the bona fide polarization markers (*Arg1*, *Mrc1*, *Nos2*) was not different in BMDM treated with PTC-EVs or CTL-EVs (Appendix A), neither was the production of TNFα, IL-6 or IL-10 (Appendix A). We could only observe a trend towards an increase in cytokines production after EV treatment, with PTC-EVs being less effective in promoting this effect compared to CTL-EVs. In conclusion, although in vivo isolated EVs targeted and impacted primary BMDMs cultured in vitro, we failed to demonstrate any reproducible effect with a global, not local, treatment of BMDMs with our in vivo isolated pool of EVs produced by different cell types.

## 4. Discussion

More and more studies suggest that extracellular vesicles (EVs) are important actors in tumor progression, as therapeutic tools, or as biomarkers, including in thyroid cancer [9,10,11,14]. However, to the best of our knowledge, no studies have examined EVs produced in vivo within the thyroid tissue and developing thyroid tumor, and thus in a complex and native environment. In this study, we used a validated genetically engineered, doxycycline-inducible mouse model of BRAF^V600E^-driven papillary thyroid cancer that mimics human PTC, and optimized an EV isolation protocol from dissociated tissue. On the one hand, we studied the heterogeneity of EV populations released within the tissue and correlated population changes with microenvironment alterations. On the other hand, we identified EV-loaded miRNAs during PTC development and established the time-course of their deregulation upon BRAF^V600E^ expression. Overall, we showed that BRAF^V600E^-induced PTC development predominantly affects epithelial EV-miRNAs which could potentially impact the immune compartment of the tumor.

The challenging isolation of EVs produced within an organ without intracellular contaminants was validated by an in-depth characterization of the high-speed pellet in accordance with the MISEV guidelines. The size of the particles and the detection of CD9, CD81 and CD63 support an enrichment of exosomes in our EV-pellets. CD9 and CD81 were found on a majority of small EVs derived from different cell lines [37,38]. We thus analyzed the cellular origin of CD9+ and CD81+ EVs by detecting the markers of the main cell types in thyroid TME: epithelial, endothelial and immune cells. EVs bearing the E-cadherin marker were considered as thyrocyte-derived EVs in normal thyroid tissue and as tumor cell-derived EVs in our model.

The large majority of thyroid- and PTC-EVs were derived from endothelial and myeloid cells. Since the thyroid gland is a densely vascularized organ [33,39], it was not so surprising to isolate a high amount of EVs derived from endothelial cells in healthy thyroids. It was still unexpected that this endothelial tissue produced much more EVs than the epithelial tissue, which is the most abundant tissue of the thyroid gland. This implied that endothelial cells are important producers of EVs, as compared to epithelial cells. Others have already studied endothelial cell-derived EVs [40,41,42] and we demonstrated an important role of these latter in developing thyroid [33]. However, to our knowledge, it has never been reported that endothelial cells could be a major source of EVs in a tissue. It was also striking to detect a large fraction of EVs derived from CD11b+ immune cells considering the very low abundance of immune cells in the control thyroid. This supports the fact that macrophages are super-producers of EVs, as proposed in [23], and that EVs are mostly bearing the CD11b marker. Immune cells also produced CD45+ EVs but the amount of this later population was relatively low. On the contrary, this CD45+ EV population doubled upon BRAF^V600E^ induction. This could be due to immune cell recruitment within the thyroid. However, recruitment did not affect the number of CD11b+ EVs. The last and minor (~2% of total) populations of CD9+/CD81+ EV produced in the thyroid were positive for the epithelial marker Ecad. However, upon BRAF^V600E^ induction, the proportion of Ecad+ EVs doubled. The BRAF^V600E^ induction and sustained activation of the MAPK pathway severely alters the gene expression and phenotype of thyrocytes which could trigger a direct effect on EV production. It is generally accepted and largely spread in the literature that cancer cells produce more EVs than normal cells [43]. The ~7-fold increased abundance of Ecad+ EVs compared to the expansion of epithelial tissue (~2-fold) supports this hypothesis. Finally, even if not the most abundant in the thyroid tumor, we are tempted to propose that these CD45+/immune cell- and Ecad+/epithelial cell-derived EVs could influence other cell types in the TME through the transfer of biologically active molecules such as miRNAs.

We elucidated the miRNome of EVs derived from control- and BRAF^V600E^-thyroids by miRNA sequencing and RT-qPCR, and provided a global signature of deregulated EV-miRNAs. A large proportion of upregulated and downregulated miRNAs identified in EVs upon BRAF^V600E^ expression have already been reported in human PTC, thereby supporting the relevance of the mouse model to study miRNAs in BRAF^V600E^-driven thyroid cancers. For example, the expression of miR-146b-5p, miR-181b-5p, miR-222-5p/3p, miR-21-5p, miR-31-5p/3p, miR-221-3p is increased in the fine needle aspiration biopsy and tissues of PTC patients, as compared to benign samples, and has further been associated with the degree of PTC tumor malignancy [44,45,46,47,48]. The mouse model allowed the dynamic analysis of miRNAs upon BRAF^V600E^ induction and revealed a primary phase of miRNAs upregulation, e.g., miR-31-5p, miR-21a-5p, miR-146b-5p and miR-21a-5p, followed by their stabilization at a later time point of tissue transformation. The rapid and sustained upregulation of these miRNAs could be inherent to their role as oncomiRs, inhibiting antagonists of the MAPK pathway. Indeed, miR-21 targets PDCD4, a MAPK pathway inhibitor, thereby enhancing proliferation and invasion, while inhibiting apoptosis in TPC-1 cell line [49,50]. In addition, the mouse model revealed at later time points the downregulation of miRNAs (e. g. miR-200 and miR-30 families) that are recognized as negative regulators of cell migration, invasion, and epithelial-to-mesenchymal transition (EMT) [51,52]. The delayed downregulation of these “cancer suppressor” miRNAs could thus promote a secondary phase in cancer progression, associated with a worsening of the phenotype, increased aggressiveness and metastasis of dedifferentiated cancer cells. EV-associated miRNAs could thus participate in these tumorigenesis processes.

The rapid and sustained upregulation of oncomiRs upon BRAF^V600E^ induction in mice thyrocytes is perfectly in line with the significant overexpression of these miRNAs in the PTC tissue of patients with BRAF mutations, as compared to another mutational status [44,53,54]. Interestingly, we found that the increased abundance of these oncomiRs in EVs isolated from mouse PTC was probably a direct consequence of their increased expression by BRAF^V600E^-induced transformed thyrocytes. Indeed, the most abundant and most deregulated EV-miRNAs were expressed by the epithelial population. We thus assume that the miRNA content of thyroid-derived EVs was affected by changes in miRNA content in BRAF^V600E^ tumor cells. The mechanisms governing miRNA loading in EVs have been recently shown to rely on small motifs in miRNA sequences. These motifs would be cell-type specific or preferentially used by certain cell types for EV loading [55,56]. Further investigations about the use of those motifs in the thyroid should confirm, or infirm, the tumor cell origin of EVs, and could then explain the impact of EVs on tumor progression and the TME. Few studies, mostly performed in vitro, reported the role of EV-miRNAs in thyroid cancer. Wu et al. demonstrated that EVs isolated from PTC cell lines increase HUVEC angiogenesis through the miR-21-5p transfer [21]. Lee et al. showed that EVs from the serum of PTC patients contain miRNA-146b and miRNA-222, and that these miRNAs increased the invasiveness of thyroid cancer cell lines [20]. It is thus conceivable that the autocrine and paracrine transfer of miRNAs via EVs are at play in thyroid cancer.

In this context, our in silico analyses highlighted a potential role for EV-miRNAs in modulating the immune microenvironment, already in the initial stages of cancer progression. Thyroid tumors have recently been classified as “inflammatory tumors” by a TCGA-based study [57]. Several studies evidenced a positive correlation between the density of tumor-associated macrophages in PTCs and larger tumors, lymph nodes metastasis, and decreased survival [58]. Tumor-associated macrophages are the most abundant and crucial non-neoplastic immune cells of the tumor stroma. TME establishment as pro- or anti-tumorigenic highly depends on macrophage polarization towards an anti-inflammatory pro-tumorigenic M2-like phenotype or a proinflammatory anti-tumorigenic M1-like phenotype, which are controlled by microenvironmental signals [59,60]. The mouse thyroid cancer model used in this study recapitulates the inflammatory nature of aggressive BRAF^V600E^ thyroid cancer with the massive recruitment of immunosuppressive macrophages expressing typical M2-like markers, such as CD206, F4/80, CD11b and Stab1 [61]. Among the more abundant EV-miRNAs identified, miR-146b-5p, miR-222-3p and miR-21a-5p have been shown to impact macrophage polarization by different mechanisms, ultimately increasing *Mrc1*, *Arg1*, and *Il10* expression and decreasing *Tnfa* and *Il12* expression, thereby promoting a M2-like phenotype [62,63]. We thus propose that Ecad+ EVs isolated from thyroid tumors could support the local establishment of a pro-tumoral immune microenvironment.

Some studies have shown the ability of tumor-derived EVs to polarize macrophages towards the M1 or M2 phenotype [62]. On the contrary, others revealed the incapacity of tumor-derived exosomes to impact on macrophage plasticity [64,65]. If the role of EVs derived from in vitro cultured cancer cell lines on macrophage polarization is still currently unclear, studying the role of an heterogenous population of EVs produced by and isolated from a tumor is a challenge. In our settings, the in vitro treatment of primary BMDMs with CTL- and PTC-derived EVs were not conclusive. The transcriptomic analysis of BMDMs treated with PTC-EVs compared to CTL-EVs pinpointed some differences compatible with an anti-inflammatory and pro-tumoral M2-like gene signature. However, the effects were marginal and variable, preventing us from reaching an unequivocal conclusion. The variability and weak effects probably come from our in vitro culture settings, highly different from the local release and action of EVs, and the heterogenous population of EVs isolated from the tumor that could trigger different effects. Indeed, in our study, we demonstrated the existence of at least three different EV-producing cell types. In addition, up to 80% of the EVs isolated from control and BRAF^V600E^ thyroid might be the same in the two conditions (number of CD31+ and CD11b+ EVs do not change upon BRAF^V600E^ induction). Conversely, since less than 10% of EVs are derived from the epithelial tumor cells, it might be difficult to observe an effect without first purifying this EV population. The difficulty to recover pure and sufficient epithelial derived-EV thus limits functional studies with in vivo-produced EVs. New technical solutions are needed to experimentally study the role of EV-miRNAs released from the tumor in M2-macrophage polarization or in the maintenance of a TME mediating pro-survival and proliferative phenotypes, thus negatively impacting patient outcomes.

## 5. Conclusions

Our work provides an in-depth characterization of in vivo-derived extracellular vesicles produced by and within normal and cancerous thyroid. This includes the identification of a global EV-microRNAs signature showing the temporal dynamic of miRNA loading in EVs during PTC development and thyrocyte dedifferentiation. The more abundant miRNAs found in tumor-derived EV and deregulated as compared to the control thyroid appear to derive from the transformed epithelial cell population. Tumor cell-derived EVs could impact on thyroid tumorigenesis and communication within the tumor microenvironment, supporting the establishment of a permissive immune microenvironment for tumor development.

## Figures and Tables

**Figure 1 biomedicines-10-00755-f001:**
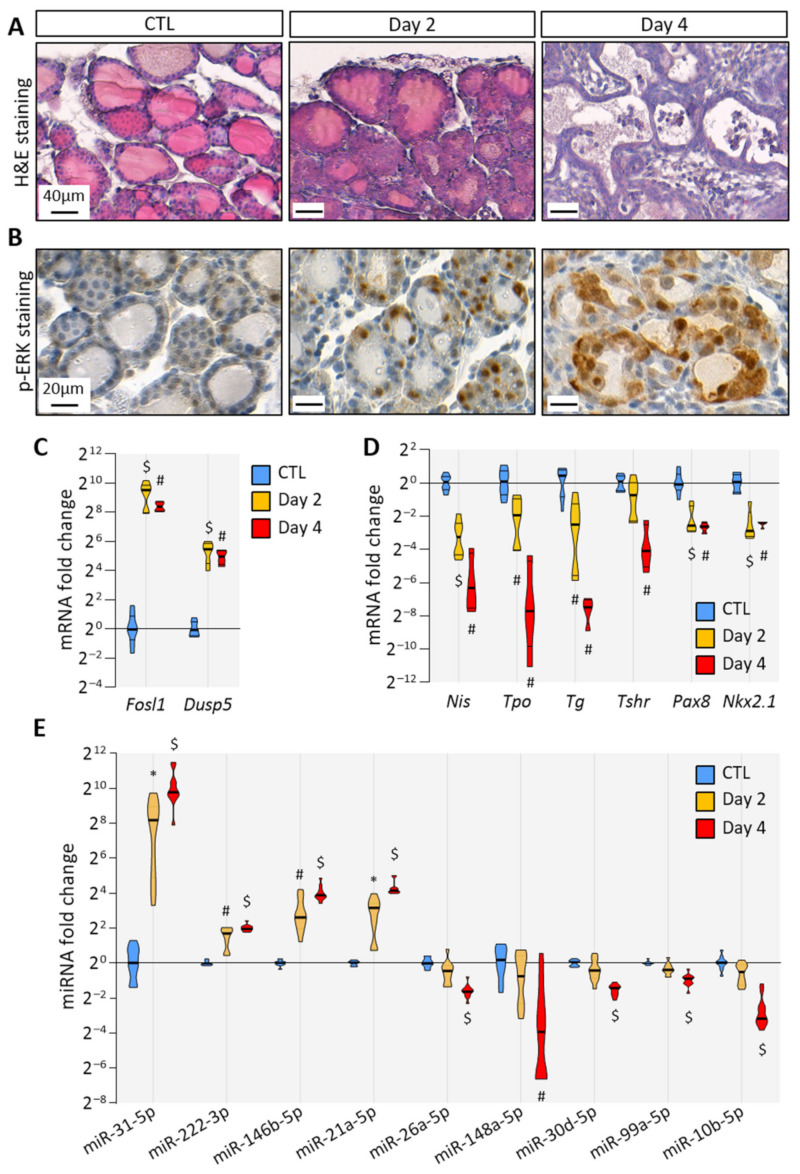
**BRAF^V600E^ expression in mouse thyrocytes triggers histological and molecular changes:** (**A**) histological staining of thyroid tissues from control and doxycycline-treated mice after 2 and 4 days of intraperitoneal injections; (**B**) immunohistochemistry of phospho-ERK (p-ERK) on thyroid tissues from control and doxycycline-treated mice after 2 and 4 days of injections; (**C**,**D**) RT-qPCR analyses of (**C**) two MAPK pathway-target genes, and (**D**) thyroid markers in thyroid tissues from control (n = 8) and doxycycline-treated mice after 2 (n = 6) and 4 days (n = 4) of injections. mRNA fold changes are normalized on the geometric mean of *Rpl27* and *Gapdh* expression and compared to the control group; (**E**) miRNA analyses of thyroid tissue from the control (n = 11) and doxycycline-treated mice after 2 (n = 10) and 4 days (n = 11) of injections. MiRNA fold changes are normalized on let-7c-5p and compared to the control group. Data are expressed as the mean ± SD (* *p* < 0.05; # *p* < 0.01; $ *p* < 0.001).

**Figure 2 biomedicines-10-00755-f002:**
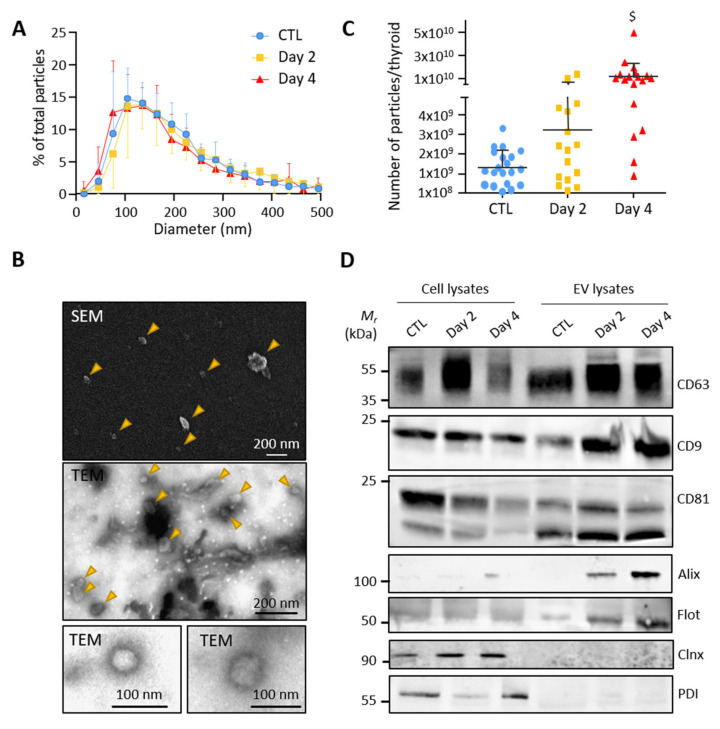
**Small EVs isolated from control and BRAF^V600E^ thyroid tissues display exosomal characteristics:** (**A**) nanoparticle tracking analysis (NTA) of resuspended EV-pellets from control and doxycycline-treated thyroids after 2 and 4 days of injections. Results are presented as the percentage of total particles as a function of particle diameter; (**B**) scanning and transmission electron microscopy of resuspended EV-pellets isolated from dissociated control thyroid. The SEM and the first TEM images show large overview of the sample; arrowheads point EV structures; (**C**) NTA quantification of particle numbers from the EV-pellets. $ stands for *p* < 0.001; (**D**) biochemical characterization of EV lysates as compared to cell lysates by the western blotting of five EV-markers (CD63, CD9, CD81, Alix and Flottilin-1) and two reticulum endoplasmic markers (Calnexin and Protein disulfide-isomerase).

**Figure 3 biomedicines-10-00755-f003:**
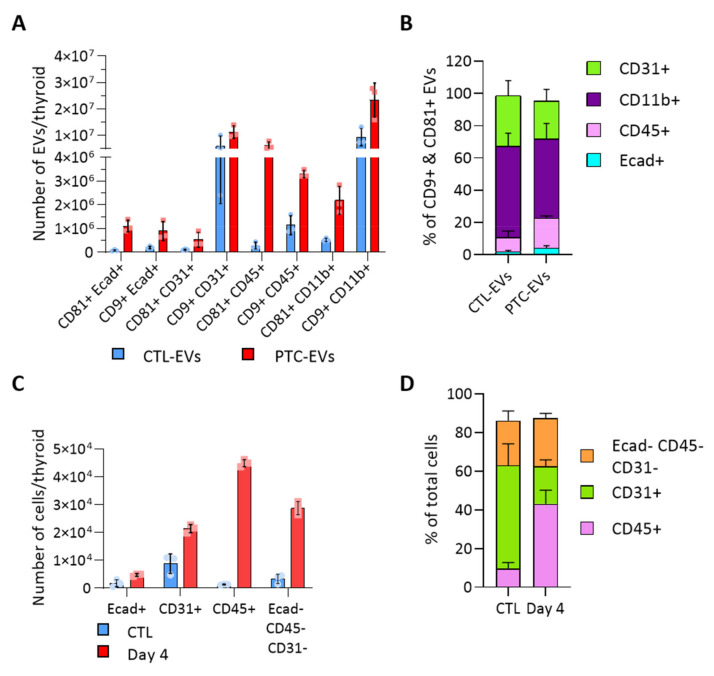
**BRAF^V600E^-driven tissue transformation alters the proportion of cellular and EV populations.** (**A**,**B**) Results obtained with the ExoView platform (n = 3). Capture antibodies were directed against CD9 and CD81 tetraspanins. Two detection panels were used with antibodies directed against Ecad and CD31 or CD45 and CD11b: (**A**) number of EVs in the control and PTC samples (day 4) according to the tetraspanin and the cellular marker present on EVs; and (**B**) percentage of EVs bearing each population marker (Ecad, CD31, CD45, CD11b) in the two, CD9+ and CD81+, EV populations. (**C**,**D**) Results obtained by FACS analysis (n = 4). Sorting antibodies were directed against Ecad, CD31 and CD45: (**C**) number of Ecad+, CD31+, CD45+ and triple negative cells in control and DOX-treated dissociated tissues (day 4); and (**D**) percentage of non-epithelial cell populations from dissociated control and DOX-treated tissues (day 4).

**Figure 4 biomedicines-10-00755-f004:**
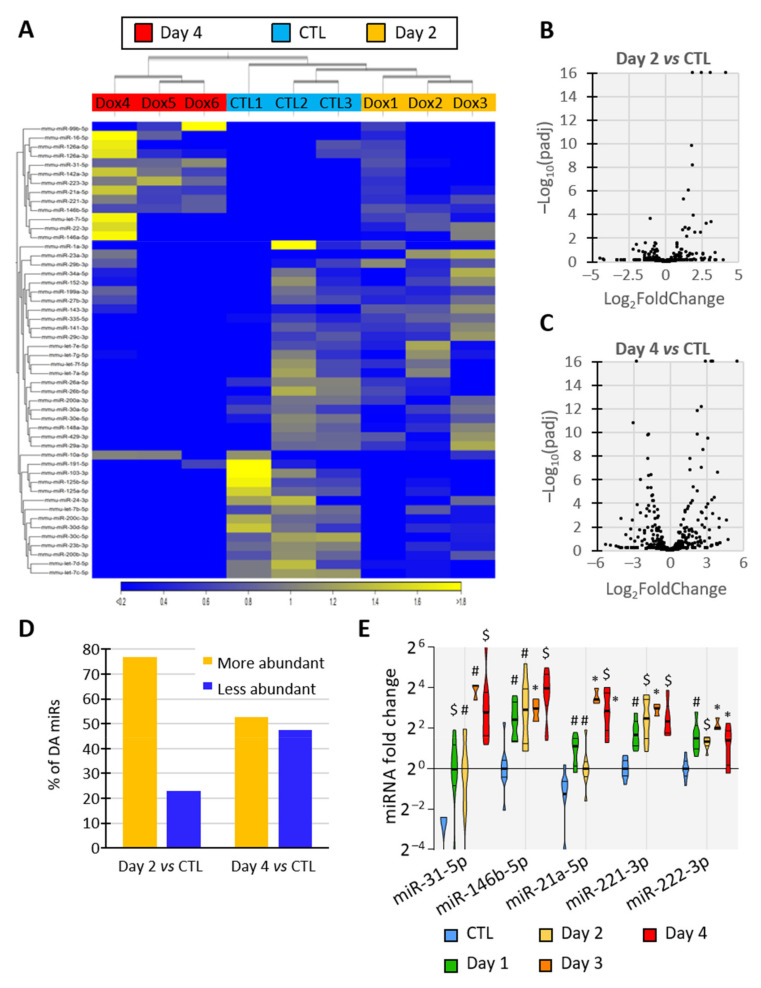
**Identity and abundance of EV-associated miRNA change with tumor progression.** (**A**–**D**) Analyses of miRNAs’ sequencing results showing miRNAs’ differential abundance in EVs from control and DOX-treated thyroids after 2 and 4 days of injections (n = 3): (**A**) unsupervised hierarchical clustering per sample and per miRNA performed on the 50 miRNAs that have the largest coefficient of variation. MiRNAs in yellow are more abundant in the corresponding tissue while miRNAs in blue are less abundant. (**B**,**C**) Volcano plot showing p-adjusted value and fold change relation in day 2 vs control (**B**) and in day 4 vs control (**C**) comparisons. (**D**) Percentage of differentially abundant (DA) miRNAs in day 2 vs control and in day 4 vs control comparisons. (**E**) RT-qPCR analyses of upregulated EV-miRNA from control and doxycycline-treated thyroids after 1, 2, 3 and 4 days of injections (n ≥ 4). MiRNA fold changes are normalized on the geometric mean of miR-126a-3p and let-7b-5p expression. Data are expressed as the mean ± SD (* *p* < 0.05; # *p* < 0.01; $ *p* < 0.001).

**Figure 5 biomedicines-10-00755-f005:**
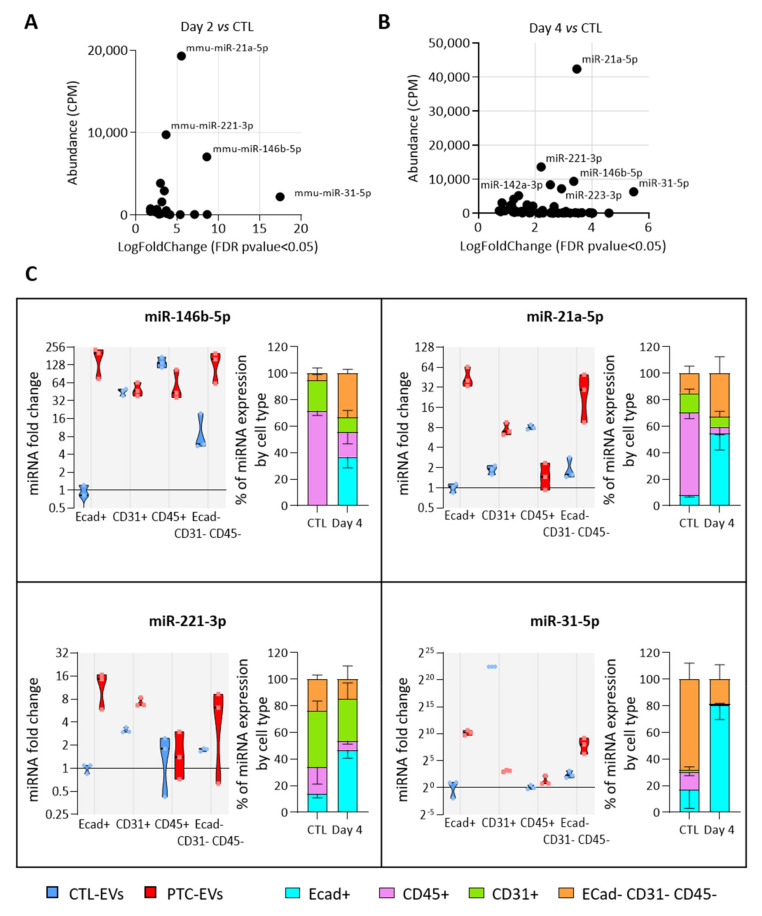
**The most deregulated and most abundant EV-miRNAs are mainly expressed and deregulated in epithelial cells.** (**A**,**B**) Scatter plots showing abundance (CPM calculated from miRNA sequencing data) of statistically deregulated EV-miRNAs between the control and DOX-treated tissues at day 2 (**A**) and day 4 (**B**). (**C**) RT-qPCR analyses of miRNA expression in FACS-sorted cell populations (n = 4). Violin plots show miRNA fold changes normalized on the abundance of a spike-in miRNA and compared to its expression in the epithelial population of control thyroids. Stacked bars show the relative expression (in percent) of miRNAs in each population of control and DOX-treated tissues. Data are expressed as the mean ± SD.

**Figure 6 biomedicines-10-00755-f006:**
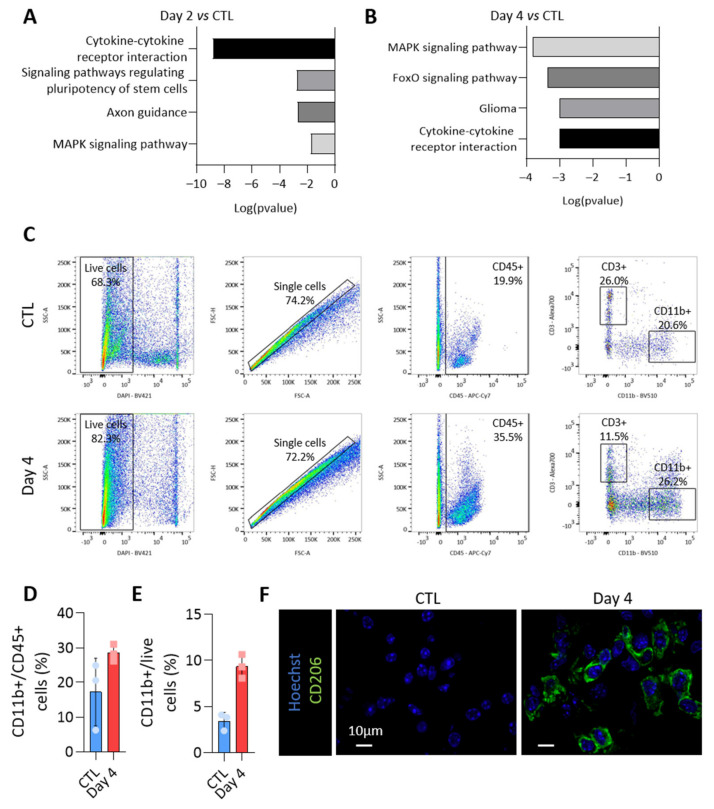
**The loading of the upregulated miRNAs in EVs could impact on the immune microenvironment of the BRAF^V600E^ tumor.** The KEGG pathway analysis using the most abundant EV-miRNAs (**A**) after 2 days and (**B**) after 4 days of BRAF^V600E^ induction, as compared to control EV-miRNAs. (**C**) Flow cytometry analyses of CD45+, CD11b+ and CD3+ immune cells from dissociated control and DOX-treated tissues (day 4). (**D**,**E**) Percentage of CD11b+ myeloid cells among the CD45+ immune population (**D**) or among the live cells (**E**). (**F**) Immunohistofluorescence of CD206 on thyroid tissues from the control and DOX-treated mice (day 4). DAPI (blue) is used to label the nuclei.

## Data Availability

Links to publicly archived datasets generated and analyzed during this study and details regarding where data supporting reported results can be found will be provided upon acceptance and publication.

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
