# Peer review of "BRAFV600E Induction in Thyrocytes Triggers Important Changes in the miRNAs Content and the Populations of Extracellular Vesicles Released in Thyroid Tumor Microenvironment"

_biomedicines, 2022, doi:10.3390/biomedicines10040755_

Round 1

Reviewer 1 Report

The manuscript is well organized and well written. The authors analyzed the cellular origin of extracellular vesicles (EVs) produced in vivo in mouse cancer model and characterized the microRNA content of these EVs.

The experimental design is very accurate, particularly when the authors demonstrate very intricate concepts.

Material and methods report all information necessary to understand the obtained results and the conclusion are well supported by the experimental data.

Author Response

We sincerely thank the reviewer for his·her positive feedback.

Reviewer 2 Report

Comment 1

The authors suggest that the tumor-derived extracellular vesicles-microRNAs could support an establishment of a permissive immune microenvironment for tumor growth upon BRAF V600E induction. The authors should add another main figure (into revised manuscript) explaining the above-mentioned hypothesis, showing interaction between miRNAs and the targets, and the recruitment of pro-tumoral macrophages and the cross-talk. 

Author Response

We sincerely thank the reviewer for his·her reviewing of our manuscript.

Our results and observations indeed suggest that EV-miRNAs could participate in the development of a permissive immune microenvironment. The data are presented in Figure 6 and Supplementary Figure 7 and are discussed in the discussion section.

This reviewer requests an additional figure “(i) explaining the hypothesis, (ii) showing interaction between miRNAs and the targets, and (iii) the recruitment of pro-tumoral macrophages”.

(i) The hypothesis is explained in the manuscript and summarized in the comment of the reviewer: “The authors suggest that the tumor-derived extracellular vesicles-microRNAs could support an establishment of a permissive immune microenvironment for tumor growth upon BRAF V600E induction”. We cannot give more details.

(ii) Showing interaction between miRNAs and the targets is out of the scope of our manuscript. In addition, some miRNA-target interactions have already been demonstrated in in vitro studies. Some examples of demonstrated regulation are cited the discussion section.

(iii) Recruitment of pro-tumoral and immunosuppressive macrophages is illustrated in Figure 6F, with the marker CD206.

Reviewer 3 Report

The authors characterized the populations of EVs from tissue of mice with an inducible mutation often related to aggressive thyroid cancer. They stratified their miRNA cargo according to the lineage of the cells that express it more abundantly. While the study is largely descriptive, the experimental design is impeccable and the article has a lot of value. Just a few questions and comments to authors:

1) the size of the EVs is larger than usual, yet I did not see any discussion on this point in the paper.

2) This reviewer did not understand figure 2B, specifically the first TEM image. It does not look clear and the authors do not really explain well what is shown. This image does not really contribute to the point of the paper as is.

3) This reviewer would like to strongly advise the authors from using unpublished work in references, since readers cannot access it. Further, the validity of this data cannot be assured due to the lack of independent peer review. The arguments constructed along this paper (line 568) should be avoided or justified using other published references. Further, some of the discussion of macrophage polarization and confounding literature regarding the effects of EVs in macrophages should be tightened for a clearer message (paragraphs starting in lines 555 and 574).

4) Please, use "day" instead of n*24h. If needed, authors can accurately describe what they mean by "day" in their materials and methods. However, the current nomenclature is confusing.

5) In materials and methods, I suggest replacing the "up-downs with p1000· by "homogenization using a 1 ml micropipette".

Author Response

We sincerely thank the reviewer for his·her positive feedback and constructive comments.

  • The size measured by NTA is effectively a bit larger than usual. However, size measurement by NTA can be criticized: it is known that NTA cannot detect particles smaller than 70 nm, thus skewing particle size distribution toward larger size. In addition, NTA cannot distinguish EVs from protein aggregates, the size of which will also be measured. This is why we also used other methods to measure EV size: (i) measurement on electron microscopy images, and (ii) of CD9+ and CD81+ EV captured on the Exoview chips. Those measurements were more compatible with classical EV size (50-80 nm).

We indeed did not discuss this point to avoid overloading this part of the Ms which is already relatively long and heavy.

  • The first TEM image in Figure 2B illustrates a larger view on the grid. MISEV guidelines and the EV-TRACK process recommend to present large fields and not only close-up images when electron microcopy experiments are presented. This allows the reader to appreciate the amount of EV in the analyzed sample. To improve understanding of Figure 2B we have now added arrowheads pointing to illustrative structures, and we give a more extensive explanation in the revised version of the Ms.

  • We agree that unpublished work might be criticized since it has not been peer-reviewed. We nevertheless consider that it is important to cite this unpublished work because it supports macrophage recruitment in the very same mouse model and with the same doxycycline-dependent BRAFV600E However, the reviewer is correct and we now refer to another published study, that support our observation, in the revised manuscript.

  • We agree that our nomenclature might be confusing. As suggested, we have replaced “n*24h” by “n day” in the revised manuscript and figures.

  • We have replaced the “bench” explanation by the more rigorous sentence proposed by the reviewer.